# Predicting Hospital Ward Admission from the Emergency Department: A Systematic Review

**DOI:** 10.3390/jpm13050849

**Published:** 2023-05-18

**Authors:** Nekane Larburu, Laiene Azkue, Jon Kerexeta

**Affiliations:** 1Vicomtech Foundation, Basque Research and Technology Alliance (BRTA), 20009 Donostia, Spain; 2Biodonostia Health Research Institute, 20014 San Sebastián, Spain; 3Biomedical Engineering Department, Mondragon Unibertsitatea, 20500 Mondragón, Spain

**Keywords:** admission risk prediction model, emergency department, patients, admission

## Abstract

Background: The emergency department (ED) is often overburdened, due to the high influx of patients and limited availability of attending physicians. This situation highlights the need for improvement in the management of, and assistance provided in the ED. A key point for this purpose is the identification of patients with the highest risk, which can be achieved using machine learning predictive models. The objective of this study is to conduct a systematic review of predictive models used to detect ward admissions from the ED. The main targets of this review are the best predictive algorithms, their predictive capacity, the studies’ quality, and the predictor variables. Methods: This review is based on PRISMA methodology. The information has been searched in PubMed, Scopus and Google Scholar databases. Quality assessment has been performed using the QUIPS tool. Results: Through the advanced search, a total of 367 articles were found, of which 14 were of interest that met the inclusion criteria. Logistic regression is the most used predictive model, achieving AUC values between 0.75–0.92. The two most used variables are the age and ED triage category. Conclusions: artificial intelligence models can contribute to improving the quality of care in the ED and reducing the burden on healthcare systems.

## 1. Introduction

The emergency department (ED) is a demanding and stressful environment for healthcare workers, due to the large number of patients and limited physicians’ availability. Overcrowding affects overall health and the functioning of the healthcare system [1]. Therefore, early identification of patients at higher risk of admission is necessary to improve outcomes and reduce the burden on healthcare systems.

In the systematic review by Maninchedda et al. [2], the main strategies and control features to reduce ED overcrowding are mentioned. The characteristics were divided into five groups: organization of work, investment in primary care, creation of new dedicated professional figures, labour, and structural modifications and implementation of predictive simulation models using mathematical algorithms. As for mathematical algorithms, in the last few years, the usage of machine learning and deep learning techniques in medicine has increased exponentially [3]. Such predictive models are clinically beneficial for understanding or identifying clinical patterns or potential risk situations in real time [4].

In the context of the ED, it is necessary to detect critical patients more accurately. The most advanced detection can be achieved with the support of artificial-intelligence (AI)-based predictive models at the time of triage. This would help in resources management and forecasting, as well as improving patient care and reducing physician overload. When looking for predictive models for ED, many studies focus on predicting the risk of ED admission. In fact, in 2010, Brabrand et al. presented a systematic review of risk score predictive models for adults admitted to the ED [5]. In 2013, Wallace et al. [6] presented a systematic review of the probability of repeat admission in community-dwelling adults, and in 2014, Wallace et al. [7] presented a systematic review of risk prediction models to predict emergency hospital admission in community-dwelling adults. In the most recent years (2021, 2022 and 2023), several systematic reviews have been presented on prediction models for admissions [8,9,10,11,12]. Hence, there are multiple studies focused on predicting patients’ risk scores when admitted to the ED or hospital.

However, to the best of our knowledge, there is no systematic review to predict ward admissions when a patient is already in the ED to manage best the patients and hospital resources in advance. Therefore, the aim of this study is to perform a systematic review to analyze the existing predictive models of ward admission from the ED. Specific objectives are: (1) to examine the variables included in ward admission prediction models, (2) to summarize the algorithms and performance of these admission predictive models; and (3) to validate the quality of the identified studies.

## 2. Methods

This review is presented according to PRISMA methodology [13]. This methodology can be summarized in four steps: search strategy, article selection, exclusion and inclusion criteria for data extraction and, if necessary, statistical analysis or quality assessment.

### 2.1. Search Strategy

To identify the research papers for the conducted systematic review we used PICO methodology [14]. Specifically, we searched the bibliographic databases of Scopus, PubMed, and Google Scholar in January of 2023. The PICO methodology involves defining the patient population (P), the intervention or exposure of interest (I), the comparison group (C), and the outcome of interest (O). By using this methodology, we were able to systematically search for articles that met our predefined criteria.

### 2.2. Study Selection

Studies were included if they met the following criteria:Population: Patients presenting to the emergency departmentIntervention: prediction models for ward admissions in the hospital from emergency department patientsComparison: not applicableOutcome: patient admission

The criteria are specified in more detail in the following Table 1, where the keywords are summarized.

After selecting the most significant words, with the aim of providing a complete overview of the available prediction models, the search was limited to 2010 to 2023, in addition to looking for articles written in English or Spanish. The search only focused on the titles, since a more general search of title, abstract, and text provided too many results, so it was decided to narrow the search by focusing on article titles. The titles of the articles did not mention the population of interest, so it was not included in the search but was later considered for the inclusion and exclusion criteria (see Section 2.3. Data Extraction). In addition, a selection of the keywords mentioned in Table 2 was created. For the search, we used the mentioned bibliographic databases with the search strings shown in Table 2.

### 2.3. Data Extraction

After automatic article selection, duplicate articles were eliminated. Next, the titles and abstracts of each of the non-duplicate articles were analyzed. Studies that did not meet the established criteria were discarded. Then we proceeded to read the selected articles, analyzing them again with the same criteria mentioned above.

Studies that met any of these criteria were excluded:Predictive models for hospital readmission.Primary population of interest focused on specific conditions (e.g., asthma or bronchiolitis).The model predicts whether a patient will be admitted in the future or requires post-triage data.Only provided risk factors evaluation and did not build a prediction model.Models that do not answer the specific question, i.e.,do not predict ward admission from the emergency department.Published in languages other than English and Spanish.Full article not available.

After selecting the studies of interest, a PRISMA diagram was built to summarize the whole process and to describe in a visual way the selection or exclusion of the studies step by step, keeping the articles of interest (see Figure 1).

### 2.4. Methodological Quality Assessment

A methodological quality assessment of the included studies was independently performed using the QUIPS tool [15] to assess the risk of bias in systematic reviews. QUIPS consists of summarizing the risks of the studies in six different domains: study participation, study attrition, prognostic factor measurement, outcome measurement, study confounding, and statistical analysis and reporting.

As a result of each of these criteria, QUIPS has a final score, named overall risk, risk of bias, or quality of evidence. It refers to the degree to which the results of a study may be reliable or generalizable to the target population. It is determined by assessing the risk of various types of biases or limitations in study design and execution. A study with a high overall risk suggests that the study results may not accurately reflect the truth, whereas a study with a low overall risk suggests that the study results are more reliable and trustworthy.

## 3. Results

In this section all the results will be presented using the methodology mentioned in the previous section.

### 3.1. Study Identification

This systematic review found a total of 367 possible articles, reducing them to 14 studies at the end of the process (Figure 1). The total of 367 articles was obtained from PubMed (n = 150), Scopus (n = 194), Google Scholar (n = 12), and other search sites other than the above (n = 11).

As seen in Figure 1, after the automatic search, the reviewer eliminated the duplicate articles, leaving 219 articles. Next, after analyzing the titles and abstracts of each of the unduplicated articles, 197 studies were discarded, and 22 studies were obtained for further analysis.

Following the same exclusion criteria, the full texts of the remaining 22 studies were analyzed and six of the 22 articles were discarded, resulting in a final 14 relevant studies. These 14 studies were examined one by one. Table 3 summarizes the most relevant data for each of them.

### 3.2. Description of Included Studies

Of the 14 final articles, three were developed in Spain [17,22,23], three in the Netherlands [18,24,27], two in Greece [19,26], two in the U.K. [21,28], two in the USA [16,20], one in France [29], and 1 in Singapore [25]. Eleven of the 14 articles included as a patient any person attending the emergency department. The remaining articles (n = 3) have some restriction on the population of the study: one excludes ED patients older than 70 years old [24]; one considers only patients older than 18 years and creates two groups of patients with the threshold of 70 years (younger and older than 70 years) [27]; and the third article [17] includes adult patients who were stable on arrival at the ED. Sample sizes ranged from 2476 [17] to 3,189,204 [22] emergency department visits. The percentage of ward admissions from the total instances ranged from 4.6% [17] to 39% [18].

The most commonly used algorithm was logistic regression, with seven out of 14 articles using it [16,17,24,25,27,28,29]. The second most used model was gradient boosting, in three studies [20,21,22]. Other studies used Random forest [19], Gaussian Naïve Bayes [26], gradient-powered decision tree modelling [18], and an artificial neural network [23]. All models predict ward admission from the ED, either giving a binary or probabilistic response.

### 3.3. Variables Used to Develop Predictive Models

Continuing with the variables used in the articles, the number of variables used varied from six variables [18] to 21 variables [16]. All the articles used hospital variables, either clinical or triage variables. In three studies, the variables were not exactly specified [18,20,21]. The following Table 4 summarizes the variables in six categories: demographics, triage information, clinical and laboratory findings, medical history, medication, and others.

As presented in Table 4, all studies used a triage data category. One study [21] emphasized that the triage category is one of the most important factors in the prediction of admission. In 11 of the 14 articles [16,17,19,21,22,23,25,26,27,28,29], a demographic category was used, and within this category, the age variable was used in all 11 studies. Four of these articles distinguish age as a good predictor of hospital admission [16,17,21,25]. Clinical and laboratory findings is the third most used category: nine articles out of 14 have used variables from that category [17,18,19,20,21,22,23,24,26,27]. Laboratory data were used extensively, with five out of nine studies using it [18,19,24,26,27]. Vital signs variables have been used in three out of nine [18,24,27] studies. In six studies, variables from the medical history have been used [21,22,23,25,27,28]. One of them determines that previous admission in the last month and previous admission in the last year are good predictors of hospital admission [21]. In one study, medications have been incorporated [17]. Finally, nine studies use other variables that do not correspond to these categories [16,19,21,23,25,26,27,28,29]. Of these unclassified variables, the mode of arrival at the emergency department is the most common one. According to one article, this variable is a good predictor [16].

### 3.4. Predictiveness of Models

The results of the 14 studies analyzed were presented using the AUC metric, which measures the separability of the classes in a classification problem (Table 5). In most of the articles, the 95% confidence interval (CI) was calculated to provide a range in which the model is estimated to perform with a 95% probability. The larger the range, the greater the uncertainty. The study with the best result was [20], conducted in the USA, which obtained an AUC of 0.92 (95% CI 0.92–0.93). In second place was article [22], conducted in Spain, which obtained an AUC of 0.8938 (95% CI 0.8929–0.8948). In third place was study [28], also conducted in the USA, which obtained an AUC of 0.8774 (95% CI 0.8752–0.8796).

### 3.5. Methodological Quality Assessment

Using the QUIPS tool, a quality assessment of the 14 studies has been carried out. Table 6 summarizes the risk level for each QUIPS section, where:

: means low risk of bias

: means moderate risk of bias
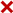
: means high risk of bias

**Table 6 jpm-13-00849-t006:** Risk of the selected studies using the QUIPS tool.

Reference	StudyParticipation	Study Attrition	Prognostic Factor	Outcome	Study Confounding	Statistical Analysis and Reporting	Overall Risk of Bias
Parker et al. [16]							
Elvira Martínez et al. [17]							
De Hond, Anne et al. [18]							
Feretzakis, Georgios et al. [19]							
Hong, Woo Suk et al. [20]							
Graham, B et al. [21]							
Cusidó, J et al. [22]							
Alexander Zlotnik et al. [23]							
A. Brink et al. [24]							
Sun, Yan et al. [25]							
Feretzakis, Georgios et al. [26]							
Lucke, Jacinta A et al. [27]							
Allan Cameron et al. [28]							
Noel, Guilhem el al. [29]							

After analyzing all the articles, the overall risk is low, since the high risk indicator does not appear in any of the articles. Figure 2 presents the risk factor of each domain for the selected articles of the review.

Most studies were able to report all QUIPS checklist items, except for the “Study Confounding” and “Study Participation” domains. This suggests that the studies included in this review had a low risk of bias, but it is important to consider the potential impact of “confounding variables” and “Study Participation”.

## 4. Discussion

This systematic review identified 14 studies that developed machine learning models for predicting hospital admissions from the ED. The number of variables used is very varied, since there are studies that use a small number of variables and others that use more than 20 variables. Most of the articles analyzed (n = 11) used data from all patients in the ED to develop their models, while a few had restrictions on the population studied. Specifically, three articles had such restrictions: one excluded patients over 70 years old [24], one focused only on patients over 18 years and created two groups based on the threshold of 70 years [27], and the third one included stable adult patients upon arrival to the ED [17]. In terms of variables, the review identified several factors that consistently predicted patient admission, such as age, mode of arrival, and the triage category. It was found that the most used models for this type of prediction are logistic regression and gradient boosting, since 10 out of 14 articles used either one of these two, although the most widely used is logistic regression, with seven out of 14 articles using it. Overall, the findings suggest that these models can be effective in accurately predicting patient ward admission, with many studies reporting good relationships between sensitivity and specificity. The article with the best result presented an AUC of 0.92 [95% CI 0.92–0.93] [20], and the article with the worst result presented an AUC of 0.766 (95% CI 0.759–0.781) [24].

Five out of the 14 articles have been published in the last three years, which shows that artificial intelligence is increasingly used in health services, more specifically in the ED.

No other systematic reviews of models predicting ward admission from the ED have been found. Therefore, this study cannot be compared with any other systematic review.

## 5. Conclusions

To date, the logistic regression algorithm is the most widely used to predict hospital ward admission from the emergency department. Although the predictive models found by this systematic review do not use environmental variables, Diaz et al. [30] observed that some environmental variables have a good relationship with ED admission. Hence, non-clinical variables should be considered when developing such models. None of the studies described implementation, and to our knowledge, none of the models are currently implemented in the ED as a prediction tool for hospital ward admission.

These predictive models improve the quality of the emergency department, including better internal management, since, by predicting the number of admissions, better resources management can be carried out.

## Figures and Tables

**Figure 1 jpm-13-00849-f001:**
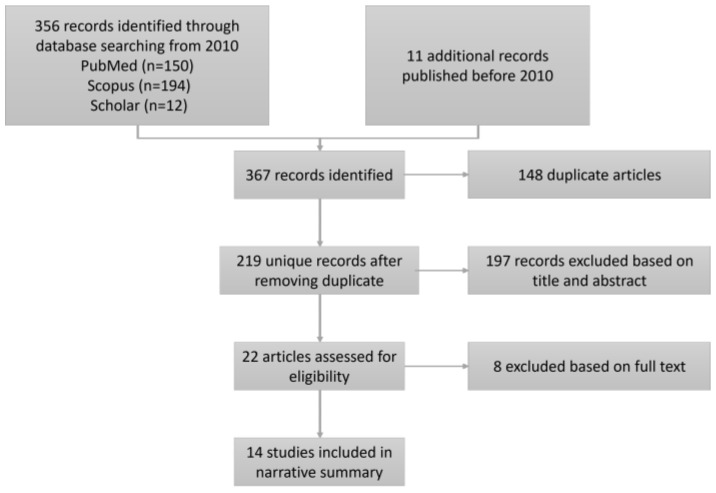
PRISMA Flow Diagram of Included Hospital Admission Prediction Models from the Emergency Department.

**Figure 2 jpm-13-00849-f002:**
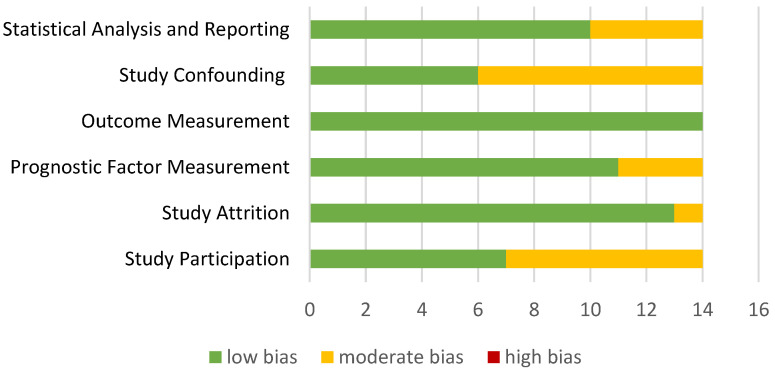
Summary of methodogical analysis.

**Table 1 jpm-13-00849-t001:** PICO methodology applied for this study.

PICO Element	Keyword Terms
*P* Population	emergency patient, emergency patients
*I* Intervention(s)	predictive models, predict, predictive, predictable, prediction, predictions, Model, models, admission, hospital admission, predictive model emergency admission, Hospital admission predictive models, Hospital admission predictive model, predictive hospital admission
*C* Comparison(s)	-
*O* Outcome(s)	Hospital admission, emergency admission, Improve triage, QoL, emergency rooms management

**Table 2 jpm-13-00849-t002:** Search strings.

	Search String	Search Date	Results
PubMed	((admission [Title]) OR admissions [Title])) AND (emergency [Title]) AND ((predict [Title]) OR (predicting [Title]) OR (prediction [Title]) OR (triage [Title]) OR (model [Title])) NOT (COVID-19)	16 January 2023	150 articles
Scopus	TITLE ((“admission” OR “admissions”) AND “emergency” AND (“predict” OR “predicting” OR “prediction” OR “triage “OR “model”)) AND NOT “COVID-19” AND PUBYEAR > 2009 AND PUBYEAR < 2023 AND (LIMIT-TO (PUBSTAGE,”final”)) AND (LIMIT-TO (SRCTYPE,”j”)) AND (LIMIT-TO (LANGUAGE,”English”) OR LIMIT-TO (LANGUAGE,”Spanish”))	16 January 2023	194 articles
Google scholar	allintitle: admission emergency hospital “machine learning”	16 January 2023	12 articles
Others	-	16 January 2023	11 articles

**Table 3 jpm-13-00849-t003:** Hospital Admission Prediction Models from the Emergency Department.

Autor	Title	Country Year	Purpose	Number of Data	Inclusion Criteria	Type of Data	Algorithm	Results
Parker, Clare Allison et al. [16]	Predicting hospital admission at the emergency department triage: A novel prediction model	USA – 2019	create a model capable of predicting the hospital admission of a patient at the time of triage	1,232,016 patients, 38.7% admitted	Emergency patients	Age group, race, zip code, day of week, time of day, triage category, mode of arrival and febrile status	Logistic regression	AUC validation: 0.825 [95% IC 0.824–0.827]
Elvira Martínez, C. M. et al. [17]	Prediction model for in-hospital admission in patients arriving in the emergency department	Spain – 2012	Hospital admission prediction model for hospital bed determination	2476 patients, of which 114 (4.6%) were admitted	Adult patients who were stable on arrival at the ED	Age, sex, triage level, initial disposition, first diagnosis, diagnostic test and medication	Logistic regression	AUC: 0.85 [95% IC 0.81–0.88; *p* < 0.001] Sensitivity: 76% Specificity: 82%
De Hond, Anne; Raven, Wouter et al. [18]	Machine learning for developing a prediction model of hospital admission of emergency department patients: Hype or hope?	Netherlands – 2021	Early identification of emergency department patients in need of hospitalization	172,104 patients, of whom 66,782 (39%) were hospitalized	Emergency patients	An increasing number of data available at triage, ∼30 min (including vital signs) and ∼2 h (including laboratory tests)	Gradient- powered decision tree modelling	AUC: 0.84 (0.77–0.88) at triage, 0.86 (0.82–0.89) at ∼30 min and 0.86 (0.74–0.93) after ∼2 h
Feretzakis, Georgios; Karlis, George; Loupelis, Evangelos et al. [19]	Using Machine Learning Techniques to Predict Hospital Admission at the Emergency Department	Greece – 2022	Develop an algorithm using ML techniques to aid clinical decision making in the emergency department	3204 patients, of which 1175 were admitted to the emergency room (36.7%)	Emergency patients	Laboratory data, age, sex, triage disposition to the emergency department and ambulance utilization	Random forest	AUC: 0.789
Hong, Woo Suk; Haimovich, Adrian Daniel; Taylor at al. [20]	Predicting hospital admission at emergency department triage using machine learning	USA – 2018	Predict hospital admission at the time of emergency department triage	560,486 patients, 29.7% were admitted	Emergency patients	Three databases: 1—triage data; 2—medical history data; 3—triage and medical history data.	Gradient boosting (XGBoost)	AUC: 0.92 [IC 95% 0.92–0.93]
Graham, B.; Bond, R.; Quinn, M.; Mulvenna, M. [21]	Using Data Mining to Predict Hospital Admissions from the Emergency Department	U.K. – 2018	Use of data mining using machine learning techniques to predict admissions to the emergency department	120,600 patients	Emergency patients	Age, mode of arrival, triage, care group, previous admission, …	Gradient boosting machine (GBM)	AUC: 0.859 Accuracy: 80.31%
Cusidó, J.; Comalrena, J.; Alavi, H.; Llunas, L. [22]	Predicting Hospital Admissions to Reduce Crowding in the Emergency Departments	Spain – 2022	Assisting in many areas of hospital administration	3,189,204 patients, 11.02% of which ended in admission	Emergency patients	Identification, cumulative visits, age, days of age, gender, CCS (Clinical Classification System), CCS frequency, classification (triage)	Gradient boosting machine (GBM)	AUC: 0.8938 [IC 95% de 0.8929–0.8948] Accuracy: 0.9113
Alexander Zlotnik, Miguel Cuchí Alfaro et al. [23]	Building a Decision Support System for Inpatient Admission Prediction with the Manchester Triage System and Administrative Check-in Variables	Spain – 2016	Give nurses the ability to allocate resources in advance using predictive modelling	255,668 patients	Emergency patients	Nine routinely collected variables routinely available right at the end of the triage process	Artificial neural network model	AUC: 0.8575 [IC 95 % 0.8540–0.8610]
A. Brink, J Alsma, H S Brink et al. [24]	Prediction admission in the older population in the Emergency Department: the CLEARED tool	Netherlands – 2020	Develop and validate a clinical prediction tool for admission to the emergency department	7606 patients	Emergency patients 70 years of age and older	Vital signs, the category of the Manchester triage system, and the need for laboratory or radiology tests	Logistic regression	AUC: 0.766 [IC 95% 0.759–0.781]
Sun, Yan; Heng, Bee Hoon; Tay, Seow Yian; Seow, Eillyne [25]	Predicting Hospital Admissions at Emergency Department Triage Using Routine Administrative Data	Singapore – 2011	Be able to predict, at the time of triage, the need for hospital admission to the emergency department	317,581 patients, of which 30.2% were admitted	Emergency patients	Demographics, ED visit or hospital admission in the previous 3 months, mode of arrival, patient acuity category (PAC) of emergency department visit and coexisting chronic diseases (diabetes, hypertension and dyslipidemia)	Logistic regression	AUC: 0.849 [IC 95% 0.847–0.851]
Feretzakis, Georgios; Sakagianni, Aikaterini et al. [26]	Predicting Hospital Admission for Emergency Department Patients: A Machine Learning Approach	Greece – 2022	Establish a machine learning model and evaluate its predictive ability for hospital admission	3204 patients	Emergency patients	Laboratory data, age, sex, use of ambulance (ambulance), triage disposition to the ED, and ED outcome (admission or discharge). (ambulance), triage disposition to ED	Gaussian NB	AUC: 0.806
Lucke, Jacinta A.; de Gelder, Jelle; et al. [27]	Early prediction of hospital admission for emergency department patients: a comparison between patients younger or older than 70 years	Netherlands – 2018	Develop models that predict hospital admissions to the emergency department	10,807 patients	Two models, 1. over 70 2. less than 70 but greater than 18	Age, sex, triage category, mode of arrival, blood test performed, chief complaint, ED visit, type of specialist, blood sample phlebotomized and vital signs	Multivariate logistic regression	AUC: <70 years: 0.86 [IC 95% 0.85 a 0.87] ≥70 years: 0.77 [IC 95% 0.75 a 0.79]
Allan Cameron, Kenneth Rodgers et al. [28]	A simple tool to predict admission at the time of triage	U.K. – 2015	Create and validate a simple clinical score to estimate the probability of admission at the time of triage	215,231 patients	Emergency patients	Triage category, age, National Early Warning Score (NEWS), ambulance arrival, referral source and admission within the last year	Mixed-effects multiple logistic model	AUC: 0.8774 [IC 95% 0.8752–0.8796]
Noel, Guilhem; Bonte, Nicolásd et al. [29]	Real-time estimation of inpatient beds required in emergency departments	France – 2019	Develop a real-time automated model to predict admissions after triage	11,653 patients, were 19.5–24.7% admitted	Emergency patients	Variables available in triage	Logistic regression	AUC: 0.815 [0.0–805.825]

**Table 4 jpm-13-00849-t004:** Variables used in each study.

Studys	Type of Variable		Variables
Parker et al. [16]	Demographics		Age group, race
	Triage information		Day of week, time of day, triage category
	Others		Zip code, febrile state, mode of arrival
Elvira Martínez et al. [17]	Demographics		Age, sex
	Triage information		Triage level, initial disposition
	Clinical and laboratory findings		First diagnosis, diagnostic test
	Medication		Medication
De Hond, Anne et al. [18]	Triage information		Data available at triage
	Clinical and laboratory findings		Vital signs, laboratory tests
Feretzakis, Georgios et al. [19]	Clinical and laboratory findings		Serum levels of urea, creatine, lactate dehydrogenase, creatine kinase, protein C-reactive, complete blood count with dialysis, paral acvated thromboplastin time, DDi-mer, Internonal normalized rao
	Demographics		Age, sex
	Triage information		Triage disposition to the emergency department
	Others		Ambulance utilization
Hong, Woo Suk et al. [20]	Clinical and laboratory findings		Clinical history data
	Triage information		Data from triage
Graham, B et al. [21]	Demographics		Age, gender
	Clinical and laboratory findings		Care group
	Triage information		Manchester triage category
	Medical history		Previous admission within the last week, month and year
	Others		Mode of arrival to hospital
Cusidó, J et al. [22]	Demographics		Identification, age, days of age, gender
	Medical history		Cumulative visits
	Triage information		Classification (triage), CCS (Clinical Classification System), CCS frequency
Alexander Zlotnik et al. [23]	Demographics		Age range, sex, insurance status
	Triage information		MTS score, MTS chief complaint group
	Medical history		ED visits (preceding 12 month)
	Clinical and laboratory findings		Visit source, visit cause
	Others		Ambulance arrival
A. Brink et al. [24]	Triage information		Manchester’s triage system category
	Clinical and laboratory findings		Body temperature, heart rate, diastolic blood pressure, systolic blood pressure, oxygen saturation, respiratory rate, baseline status, the need for laboratory or radiology testing
Sun, Yan et al. [25]	Demographics		Age, gender and ethnicity
	Triage information		Patient acuity category (PAC) of emergency room visit
	Medical history		ED visit or hospital admission in the previous 3 months
	Others		Mode of arrival, coexisting chronic diseases (diabetes, hypertension and dyslipidemia)
Feretzakis, Georgios et al. [26]	Demographics		Age, sex
	Triage information		ED triage disposition
	Clinical and laboratory findings		Serum levels of Urea (UREA), creatinine (CREA), lactate dehydrogenase (LDH), creatine kinase (CPK), C-reactive protein (CRP), complete blood count with differential, including leukocytes, white blood cells (WBC) and white blood cells (WBC), neutrophil count (NEUT%), lymphocyte count (LYM%), hemoglobin (HGB) and platelets (PLT), activated partial thromboplastin time (aPTT), thrombocyte (aPTT), D-dimer, international normalized ratio (INR)
	Others		Ambulance use (ambulance)
Lucke, Jacinta A et al. [27]	<70 years	Demographics	Age, sex
		Triage information	Triage category, chief complaint
		Clinical and laboratory findings	Blood test performance, all vital signs, phlebotomized blood sample
		Medical history	ED revisit
		Others	Mode of arrival, type of specialist
	≥70 years	Demographics	Age
		Triage information	Triage category, chief complaint
		Clinical and laboratory findings	Performance of blood work, phlebotomized blood sample, all vital signs except heart rate
		Medical history	ED visits
		Others	Mode of arrival
Allan Cameron et al. [28]	Demographics		Age
	Triage information		Triage category, National Early Warning Score (NEWS)
	Medical history		Admission in the last year
	Others		Ambulance arrival, referral source
Noel, Guilhem et al. [29]	Demographics		Sex, age, age category
	Triage information		Triage category, final diagnosis
	Others		Mode of arrival

**Table 5 jpm-13-00849-t005:** Summarization of the results.

Reference	Result: AUC
Hong, Woo Suk et al. [20]	0.92 (95% CI 0.92–0.93)
Cusidó, J et al. [22]	0.8938 (95% CI 0.8929–0.8948)
Allan Cameron et al. [28]	0.8774 (95% CI 0.8752–0.8796)
Elvira Martínez et al. [17]	0.85 (95% CI 0.81–0.88)
Sun, Yan et al. [25]	0.849 (95% CI 0.847–0.851)
Graham, B et al. [21]	0.859
Parker et al. [16]	0.825 (95% CI 0.824–0.827)
Alexander Zlotnik et al. [23]	0.8575 (95% CI 0.8540–0.8610)
De Hond, Anne et al. [18]	0.86 (0.77–0.88) at triage 0.86 (0.82–0.89) at 30 min 0.86 (0.74–0.93) after approximately 2 h
Noel, Guilhem el al. [29]	0.815 (0.805–0.825)
Feretzakis, Georgios et al. [26]	0.806
Feretzakis, Georgios et al. [19]	0.789
Lucke, Jacinta A et al. [27]	0.86 (95% CI 0.85–0.87) for those older than 70 years 0.77 (95% CI 0.75–0.79) for those younger than 70 years
A. Brink et al. [24]	0.766 (95% CI 0.759–0.781)

## Data Availability

Not applicable.

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
