# Peer review of "Predicting Hospital Ward Admission from the Emergency Department: A Systematic Review"

_jpm, 2023, doi:10.3390/jpm13050849_

Round 1

Reviewer 1 Report

The topic of this paper is a hot topic. However, I have some suggestions to improve the paper.

1. Introduction

For an international journal, do not describe the process in your department and do not refer to Spanish news.

2. Methods

This section describes methods that are well known and only need to be referred to. (PICO line 62-64; quips line 115-127

3. Results

We get the data twice, in a table and in a text. Highlight only the important aspects in the text

5. Conclusions

You may only make conclusions that are supported by data. verbs as "could" indirectly predict; it is "believed" that is a statement that your data do not support your conclusion. This is not scientific language. Your study examined prediction models of admission, thus your conclusions should be limited to the value of these predicition models.

Author Response

Dear reviewer,

The authors are very indebted to the Reviewers, for their detailed, inspiring, and constructive criticism. In the revised version, we have addressed all the comments by Reviewers, thus achieving a substantially improved version of our paper.

In the following, we briefly summarize the main improvements of the first version. Then, we will address one-by-one all the points risen by Reviewers (in blue).

MAIN CHANGES

On the basis of the Reviewers’ comments, we have improved our original submission along several main directions.

  1. We have changed the introduction
    1. Clearer and more readable description
    2. better overview
  2. Reduction of the results
    1. To avoid redundancy
  3. We have changed the conclusion
    1. Scientific language and objective conclusions from the studied papers

One-by-one analysis Reviewers’ comments

Black text: reviewer’s comment

Blue text: authors’ response

  1. Introduction: For an international journal, do not describe the process in your department and do not refer to Spanish news.

Thank you for the comment. We agree that the use of Spanish news and the description of a specific department is inappropriate. Therefore, we have changed the focus of the introduction to have more general vision on the situation worldwide. (see lines 61-65 in the current version with track changes)

Besides, we have considered important to have a more detailed description on the studies carried so far in the field, specially looking to other systematic reviews, and compare them to the approach of our paper (see lines 66-85 in the current version with track changes)

  1. Methods: This section describes methods that are well known and only need to be referred to. (PICO line 62-64; quips line 115-127

Thank you for your comment. We have reduced the description of PICO (see lines 116-118 in the current version with track changes).

In addition to it, we have removed the description of QUIPS in lines 115 to 127 (see lines 169-185 in the current version with track changes).

  1. Results: We get the data twice, in a table and in a text. Highlight only the important aspects in the text

You are right. We have reduced the part of the results of the predictive models as there was too much redundancy in the table and text. So, the papers with the best three results have been described and the others are expressed in the summary table – Table 5 (see lines 264-275 in the current version with track changes).

  1. Conclusions: You may only make conclusions that are supported by data. verbs as "could" indirectly predict; it is "believed" that is a statement that your data do not support your conclusion. This is not scientific language. Your study examined prediction models of admission, thus your conclusions should be limited to the value of these prediction models.

Thank you for your comment. We change the conclusions using scientific language and limiting to the value of the studies found (see conclusions in the current version with track changes).

Please, kindly let us know whether the response is adequate, since we are happy to follow any further instructions you may want to give us.

Thank you again for your time.

Best Regards,

Nekane Larburu, 14th April 2023

Reviewer 2 Report

The article doesn 't add valuable informations on the topic declared.

More details on variables analyzed and on clinical implications of  using AI models in triage should be described.

249 correggere "triaje"

Author Response

Dear reviewer,

The authors are very indebted to the Reviewers, for their detailed, inspiring, and constructive criticism. In the revised version, we have addressed all the comments by Reviewers, thus achieving a substantially improved version of our paper.

In the following, we briefly summarize the main improvements of the first version. Then, we will address one-by-one all the points risen by Reviewers (in blue).

MAIN CHANGES

On the basis of the Reviewers’ comments, we have improved our original submission along several main directions.

  1. We have changed the introduction
    1. Clearer and more readable description
    2. better overview
  2. Reduction of the results
    1. To avoid redundancy
  3. We have changed the conclusion
    1. Scientific language and objective conclusions from the studied papers

One-by-one analysis Reviewers’ comments

Black text: reviewer’s comment

Blue text: authors’ response

  1. The article doesn't add valuable informations on the topic declared.

Thank you for the comment. It has helped us to realize that probably the main added value of the paper was not clear. Therefore, to make clearer the scope of the paper and the difference with other studies, we have described in the introduction section the approach of other studies (especially other systematic reviews in the field) and the added value of our study, which is focused on the risk prediction of ward admission when a patient is already in the emergency department. (see lines 73-88 in the current version with track changes).

  1. More details on variables analyzed and on clinical implications of using AI models in triage should be described.

Thank you for the comment. Section 3.3. presents the variables used in the studied papers and provides the overall view on the most frequent variables used for this type of predictive models. Please, kindly let us know whether this is sufficient.

Regarding the implications of using AI models in triage, we have changed the focus of the introduction to present best the implications of AI (see lines 47-60 in the current version with track changes).

  1. 249 correggere "triaje"

You are right. we have changed the spelling mistake.

Please, kindly let us know whether the response is adequate, since we are happy to follow any further instructions you may want to give us.

Thank you again for your time.

Best Regards,

Nekane Larburu, 14th April 2023

Round 2

Reviewer 1 Report

perfect adaptation of the paper

Reviewer 2 Report

English could be improved and check some errors (airse instead of arise....)